# *Edwardsiella tarda* TraT is an anti-complement factor and a cellular infection promoter

Mofei Li[1,2], Meng Wu[1,3,4], Yuanyuan Sun[1,3] & Li Sun [1,3,4 ✉]

*Edwardsiella tarda* is a well-known bacterial pathogen with a broad range of host, including fish, amphibians, and mammals. One eminent virulence feature of *E. tarda* is its strong ability to resist the killing of host serum complement, but the involving mechanism is unclear. In this report, we identified *E. tarda* TraT as a key player in both complement resistance and cellular invasion. TraT, a surface-localized protein, bound and recruited complement factor H onto *E. tarda*, whereby inhibiting complement activation via the alternative pathway. TraT also interacted with host CD46 in a specific complement control protein domain-dependent manner, whereby facilitating the cellular infection and tissue dissemination of *E. tarda*. Thus, by acting as an anti-complement factor and a cellular infection promoter, TraT makes an important contribution to the complement evasion and systemic infection of *E. tarda*. These results add insights into the pathogen-host interaction mechanism during *E. tarda* infection.

[1] CAS & Shandong Province Key Laboratory of Experimental Marine Biology, Institute of Oceanology, Center for Ocean Mega-Science, Chinese Academy of Sciences, Qingdao 266071, China. [2] Tianjin Key Laboratory of Animal and Plant Resistance, College of Life Sciences, Tianjin Normal University, 393 West Binshui Road, Xiqing District, Tianjin 300387, China. [3] Laboratory for Marine Biology and Biotechnology, Pilot National Laboratory for Marine Science and Technology (Qingdao), Qingdao, China. [4] College of Earth and Planetary Sciences, University of Chinese Academy of Sciences, Beijing 100049, China. ✉email: lsun@qdio.ac.cn

*E*dwardsiella tarda is a Gram-negative bacterium and a well-known pathogen that causes systemic infection in a broad range of animal hosts, including fish, reptiles, birds, amphibians, and mammals[1]. E. tarda can replicate intracellularly in host phagocytic cells and evade the killing of serum complement[2–5]. Serum can enhance the tricarboxylic acid cycle of E. tarda, which decreases the formation of membrane attack complex (MAC) on bacterial cells, resulting in serum resistance[6]. Virulence-related factors, such as Sip1 and 2[7,8], Eta1[9], Inv1[10], and the lysozyme inhibitors MliC and Ivy[11,12], are involved in serum survival. However, the key factors required for E. tarda serum resistance still remain to be discovered.

Complement is a central part of the innate immunity and plays a vital role in defense against pathogens[13]. The complement system is activated via three pathways, i.e., the classical, the alternative, and the lectin pathways, all which produce the C3 convertase that cleaves C3 into the C3a and C3b fragments[13,14]. C3b associates with the C3 convertase to form the C5 convertase, which cleaves C5 into the C5a and C5b fragments[15]. C5b assembles together with C6, C7, C8, and C9 to generate the MAC that induces osmotic lysis of the target cells[15,16]. The complement system has a complex and strict regulation mechanism to prevent excessive complement activation and tissue damage[15]. The regulatory system contains soluble molecules and membrane proteins[17]. The soluble molecules include complement factor I (CFI), complement factor H (CFH), CFH-related proteins, C1 inhibitor, and C4 binding protein[17]. The membrane proteins include decay-accelerating factor (DAF), CD46, CD59, and complement receptor 1 (CR1)[17]. CFH and CD46 act as cofactors in CFI-mediated cleavage of C3b, which generates C3c and C3d, thus preventing complement activation and amplification[18,19].

Pathogens have evolved various mechanisms to inhibit complement activation and survive in serum[14,20,21]. These mechanisms include the presence of a capsule against complement activation, recruitment of host complement negative regulators, expression of complement negative regulators-like proteins that cleave complement proteins on the bacterial surface, and secretion of proteases that inactivate complement proteins[14,20,21]. Many specific molecules expressed by pathogens against the complement system have been reported[22–28]. TraT is a cell surface-exposed, outer membrane lipoprotein involved in surface exclusion[29,30]. In Escherichia coli K12, overexpression of traT increased bacterial survival in serum[31,32], and the presence of TraT decreased C3 deposition on bacteria[33]. TraT inhibited complement lysis of sensitized erythrocytes mainly by inhibiting the action on C6 activation or C5b6 complex formation[34]. In Salmonella typhimurium, TraT also contributed to serum resistance[35]. However, the molecular mechanism of TraT-associated serum resistance remains to be investigated.

In previous studies, we found that E. tarda resists serum killing by preventing complement activation via the alternative pathway[5]. In this study, we reported the identification of E. tarda TraT as a key player in the serum survival of E. tarda by acting as a receptor for CFH and inhibiting complement activation. In addition, TraT also serves as a ligand for host cell CD46 and contributes to E. tarda invasion into host cells and tissues.

## Results
**Identification of *traT* as an essential gene for *E. tarda* to resist serum killing by blocking complement activation**. When treated with mouse serum, E. tarda TX01 exhibited a survival rate of $87 \pm 12\%$, which was in sharp contrast to E. coli DH5α, a serum-sensitive strain[5], that exhibited no serum survival at all. Preliminary studies indicated that TX01 bound to C3 and other complement factors (see below section). To search for the

bacterial surface proteins potentially involved in complement interaction, a series of isogenic mutants of TX01 were created, each bearing a markerless deletion of a putative outer membrane protein gene. One of the mutants, TX01ΔtraT, in which the traT gene was deleted, displayed markedly reduced serum survival (Fig. 1a). Introducing traT back into TX01ΔtraT, which resulted in the traT complement strain TX01ΔtraT/traT, restored the serum resistance ability (Fig. 1a). TEM showed that serum treatment damaged the cellular structure of TX01ΔtraT, but not that of TX01ΔtraT/traT or TX01 (Fig. 1b). The remaining complement activity in bacteria-incubated serum was determined by measuring the hemolytic and bactericidal activities of the serum, which showed that both activities in TX01ΔtraT-incubated serum were significantly ($P < 0.05$) lower than that in TX01- or TX01ΔtraT/traT-incubated serum (Fig. 1c, d). Consistently, the amount of C5a and the chemotactic activity in TX01ΔtraT-treated serum were significantly ($P < 0.05$) higher than that in TX01-treated serum (Fig. 1e, f). These results indicate that TX01 resists serum killing by blocking complement activation in a manner that depends essentially on *traT*.

**traT is required for *E. tarda* to interact with complement factor H and other complement components**. To examine the mechanism whereby *traT* functions in complement inhibition, we determined the effect of *traT* on complement deposition on the surface of E. tarda. Following incubation with serum, C3c was detected on the surface of TX01, TX01ΔtraT, and TX01ΔtraT/traT, as well as E. coli (Fig. 2a); C5 β chain and the cleaved CFB fragment, Bb (~70 kDa), were detected on TX01ΔtraT and E. coli, but not on TX01 or TX01ΔtraT/traT (Fig. 2b, c). CFH and CFI were detected abundantly on TX01 and TX01ΔtraT/traT, but scantily on TX01ΔtraT and absent on E. coli (Fig. 2d–f). In agreement with this observation, compared to TX01-treated serum, which had relatively little CFH left, TX01ΔtraT-treated serum retained much more free CFH (Fig. 2g), indicating a defected ability of TX01ΔtraT to bind and withhold CFH. A similar defect of TX01ΔtraT was also observed in its binding to purified CFH protein (Fig. 2h, i). Unlike TX01, which bound purified CFI in the presence of CFH, TX01ΔtraT failed to bind CFI regardless of the presence of CFH (Fig. 2j). When CFH or CFI was depleted from serum, the serum survival rates of TX01 and TX01ΔtraT/traT, but not TX01ΔtraT, were significantly ($P < 0.05$) reduced (Fig. 2k). Taken together, these results indicate that *traT* is required for interaction with CFH, which is essential to E. tarda survival in serum.

**Recombinant TraT (rTraT) binds complement factor H and inhibits complement activation in a structure-dependent manner**. We next examined whether TraT could serve as a binding receptor for CFH. TraT is composed of 245 amino acid residues, with a predicted molecular mass of 26.2 kDa and a predicted pI of 9.0. Its function in E. tarda is unknown. Microscopy detected TraT on the surface of E. tarda (Supplementary Fig. 1). Following incubation with serum, rTraT was co-immunoprecipitated with CFH (Fig. 3a), suggesting a direct interaction between rTraT and serum CFH, which supported the observations with TX01ΔtraT in the above sections. The binding of TraT to CFH was further confirmed by using purified CFH protein (Fig. 3b) and surface plasmon resonance (SPR) analysis, which showed specific binding of rTraT to CFH, with a $K_D$ value of 3.15 μM (Supplementary Fig. 2). When rTraT was pre-incubated with serum, it significantly ($P < 0.01$) reduced the hemolytic and bactericidal activities of the serum (Fig. 3c, d), indicating a negative effect of rTraT on complement activation. Structure modeling revealed five serial helices (α1–α5) in TraT

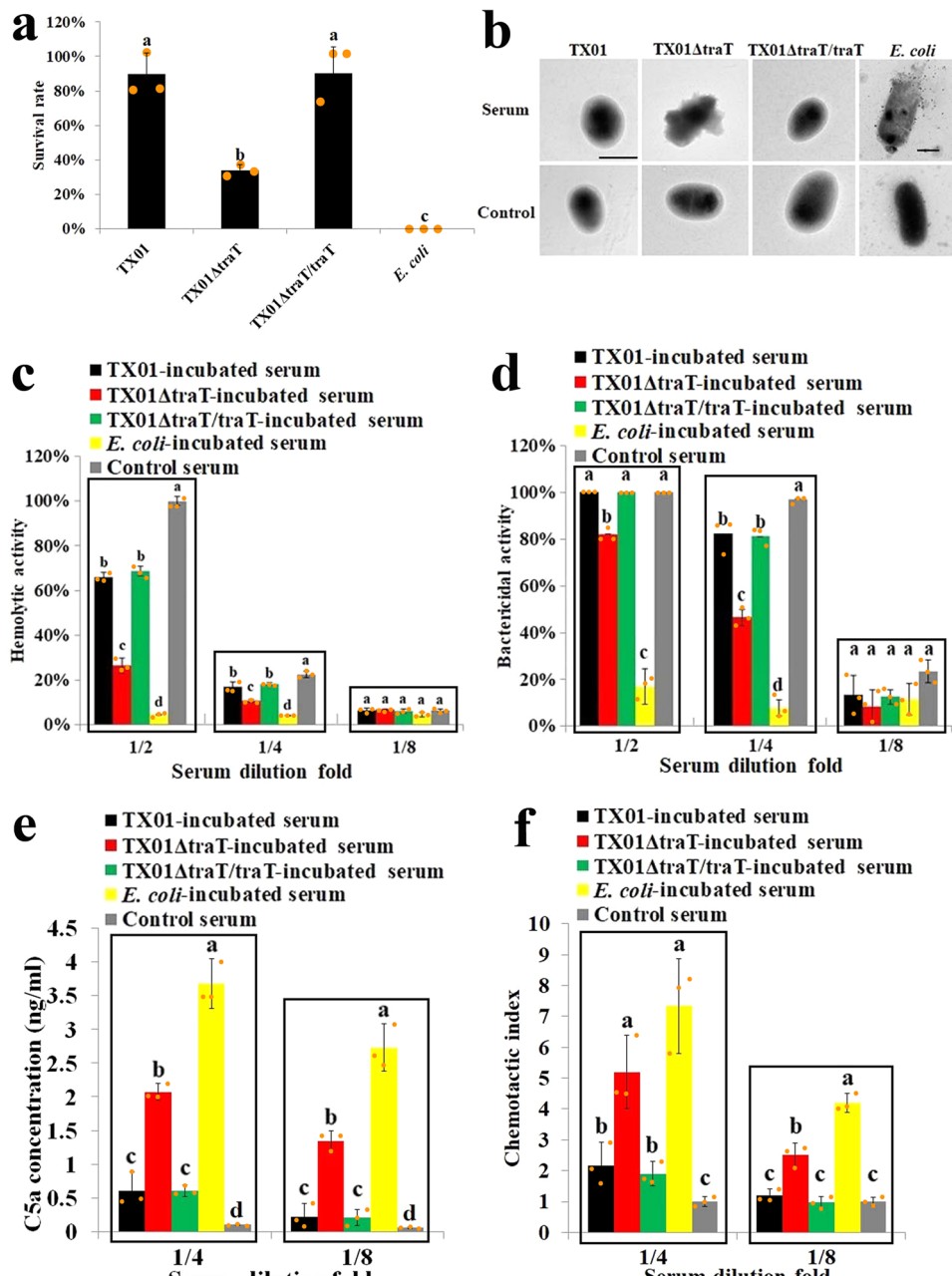

**Fig. 1 *Edwardsiella tarda* TraT is essential to serum resistance and complement activation. a** *E. tarda* wild type (TX01), *traT* mutant (TX01ΔtraT), and *traT* complement (TX01ΔtraT/traT) strains were treated with serum for 1 h, and bacterial survival was then determined. *Escherichia coli* was included for comparison. Different letters indicate statistical significance among the samples, *P* < 0.05. **b** *E. tarda* variants and *E. coli* were treated with or without (control) serum as above and subjected to transmission electron microscopy. Bar, 1 μm. **c–f** *E. tarda* variants and *E. coli* were incubated with diluted serum for 1 h. The hemolytic activity (**c**), bactericidal activity (**d**), C5a concentration (**e**), and chemotactic activity (**f**) in the serum were determined. In all panels except (**b**), data are the means of three experiments and shown as means ± SEM. For panels **c**, **d**, **e**, and **f**, in each serum dilution, different letters indicate statistical significance among the samples, *P* < 0.05.

(Fig. 3e). To examine the functional importance of these helices, four mutants of TraT, i.e., M1, M2, M3, and M4, were created, which contain only the first one helix (α1), the first two helices (α1–α2), the first three helices (α1–α3), and the first four helices (α1–α4), respectively (Fig. 3e). When expressed in *E. coli* (Supplementary Fig. 3), rTraT and the M4 mutant, but not M1, M2, or M3, significantly (*P* < 0.01) promoted the serum survival of *E. coli* (Fig. 3f). Furthermore, the heterologous TraT and M4 were able to recruit CFH onto *E. coli* (Fig. 3g). These results indicate that *E. tarda* TraT is able to interact directly with CFH, probably in a manner that depends on the first four helices of TraT.

**TraT plays a vital part in *E. tarda* infectivity**. Since, as shown above, TraT is essential for *E. tarda* to evade complement-mediated killing, we further examined its contribution to the overall pathogenicity of *E. tarda*. *traT* knockout did not affect the growth of *E. tarda* in LB medium or under oxidative condition (LB medium supplemented with 1 mM H₂O₂) (Supplementary Fig. 4), but had a significant (*P* < 0.05) effect on the infectivity of *E. tarda* (Fig. 4). In vitro and in vivo infection studies showed that compared to TX01 or TX01ΔtraT/traT, TX01ΔtraT was significantly (*P* < 0.01) impaired in the ability to infect mouse macrophages and induce cell death (Fig. 4a, b), and to

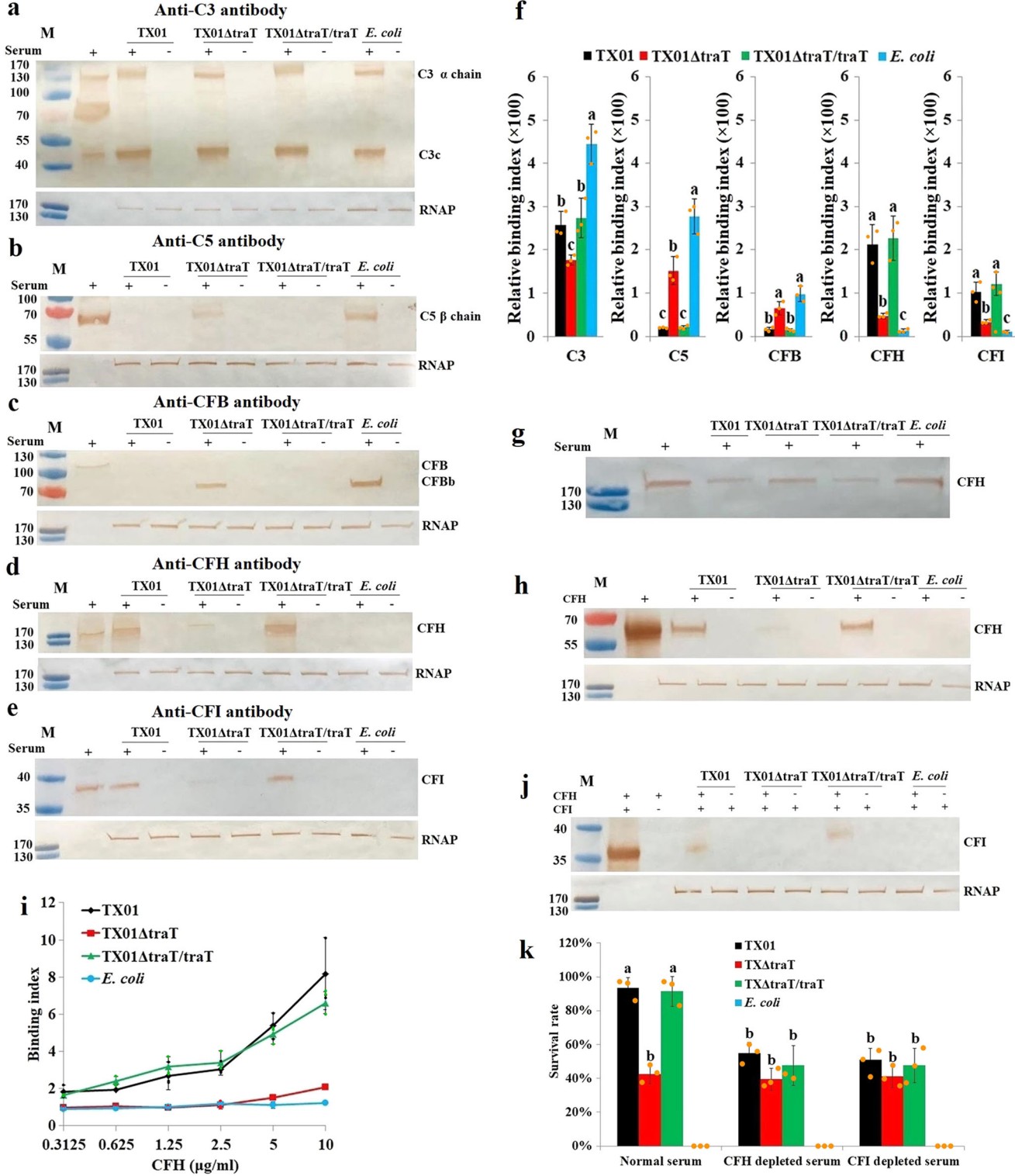

disseminate in the blood, intestine, spleen, and kidney of the mice after infection via tail vein injection (Fig. 4c). When inoculated into mice at the dose of $5 \times 10^7$ CFU, TX01 and TX01ΔtraT/traT caused 100% mortality, whereas TX01ΔtraT caused no mortality at all. Histopathology of the infected mice showed that, while TX01 and TX01ΔtraT/traT induced marked reduction of spleen lymphocytes and abscission of intestinal epithelial cells and connective tissue hyperplasia in the lamina propria, TX01ΔtraT induced no apparent histological abnormality in host tissues (Fig. 4d). Examination of inflammatory gene expression showed

that compared to TX01 or TX01ΔtraT/traT, TX01ΔtraT induced significantly ($P < 0.05$) lower inductions of interleukin (IL)-1β, IL-6, IL-18, IL-27, IL-33, tumor necrosis factor (TNF) α, and CXCL2 in intestine, and significantly ($P < 0.05$) lower inductions of IL-6, IL-27, and CXCL2 in spleen (Fig. 4e), suggesting an involvement of TraT, in a tissue-dependent manner, in the inflammatory response induced by *E. tarda*.

**Identification of CD46 as a host cell receptor for TraT.** Since TraT is localized on the surface of *E. tarda* and essential to

**Fig. 2 traT is required for Edwardsiella tarda-complement interaction. a–f** Interaction of *E. tarda* TX01 variants with serum complement components. TX01 variants and *Escherichia coli* were incubated with serum for 1 h, and western blot was performed to detect bacteria-bound serum C3 (**a**), C5 (**b**), CFB (**c**), CFH (**d**), and CFI (**e**). RNA polymerase β (RNAP) was used as a loading control. In **f**, the above serum-treated bacteria were incubated with the antibody against C3, C5, CFB, CFH, or CFI and then with FITC-labeled secondary antibody. The amount of the bacteria-bound antibody was determined by measuring fluorescence intensity. **g** The CFH left in the serum incubated with *E. tarda* TX01 variants. TX01 variants and *E. coli* were incubated with serum for 1 h; the remaining CFH in the serum was detected by western blot. **h, i** Binding of *E. tarda* TX01 variants to purified CFH. TX01 variants and *E. coli* were incubated with or without purified CFH for 1 h, and the bacteria-bound CFH was determined by western blot (**h**) and ELISA (**i**). RNAP, a loading control for western blot. **j** Requirement of CFH for the binding of *E. tarda* TX01 variants to CFI. TX01 variants and *E. coli* were incubated with or without purified CFI in the presence or absence of purified CFH for 1 h. Bacteria-bound CFI was determined by western blot. RNAP, a loading control. **k** Effect of CFH and CFI on the serum survival of *E. tarda* TX01 variants. TX01 variants were incubated with normal serum, CFH-depleted serum, or CFI-depleted serum for 1 h, and bacterial survival was determined. In all western blots, lane M represents molecular weight markers. In panels **f, k**, the data are the means of three experiments and shown as means ± SEM. Different letters indicate statistical significance among the samples, *P* < 0.05.

cellular invasion, we examined its potential interaction with host cells. Following incubation with RAW264.7 cells, rTraT was detected on the cells (Fig. 5a), indicating a cellular binding capacity of rTraT, which, however, was significantly (*P* < 0.01) reduced when rTraT was pre-treated with CFH (Fig. 5b). Furthermore, pre-treatment of RAW264.7 cells with rTraT significantly (*P* < 0.01) protected the cells against the invasion of TX01 (Fig. 5c), suggesting an importance of TraT-mediated host cell binding in the cellular invasion of *E. tarda*. These observations led us to search for the host cell protein that may serve as a receptor for TraT. Since CD46 is known to be an outer membrane protein involved in complement regulation[36], we examined whether CD46 could be a binding target for TraT. For this purpose, mouse CD46 (mCD46) was expressed in the human HEK293T cells (Supplementary Fig. 5). rTraT was found to be co-immunoprecipated with the mCD46 in mCD46-expressing HEK293T cells (Fig. 5d), suggesting a direct interaction between rTraT and mCD46. Infection study showed that heterologous expression of mCD46 in HEK293T cells significantly (*P* < 0.05) enhanced the cellular invasion of TX01 and TX01ΔtraT/traT, but not TX01ΔtraT (Fig. 5e). In line with this result, when *E. tarda* infection was performed with the mouse cells RAW264.7, which naturally express mCD46, the cellular invasions of TX01 and TX01ΔTraT/TraT at 1, 2, and 4 h post-infection (hpi) were all significantly (*P* < 0.05) reduced by anti-mCD46 antibody, whereas the invasions of TX01ΔTraT were unaffected by the antibody at 1 and 4 hpi (Fig. 5f).

**mCD46 interacts with *E. tarda* in a specific complement control protein (CCP)-dependent manner.** Given the importance of CD46 in *E. tarda* infection, we further investigated the interaction between CD46 and *E. tarda*. We found that recombinant mCD46 (rmCD46) bound TX01 (Fig. 6a), and the bound rmCD46 decreased the binding of TX01 to mCD46-expressing HEK293T cells (Fig. 6b). Similar inhibition of TX01 attachment to the mouse RAW264.7 cells by pre-bound rmCD46 was also observed (Supplementary Fig. 6). Since CD46 is known to be a receptor for CFI and C3b[19], we examined whether the binding of TX01 affected the ability of CD46 to bind C3b or CFI. We found that the presence of TX01 had no significant effect on the binding of rmCD46 to C3b or CFI (Supplementary Fig. 7), suggesting that TX01 and C3b/CFI interacted with mCD46 at different sites. Since mCD46 contains four complement control protein (CCP) domains in the extracellular region, we examined their potential importance in the functioning of mCD46. For this purpose, six mCD46 mutants, i.e., mCD46ΔCCP4, mCD46ΔCCP3, mCD46ΔCCP2, mCD46ΔCCP1, mCD46ΔCCP34, and mCD46ΔCCP12, were constructed, which bear a deletion of the 4th, 3rd, 2nd, 1st, the 3rd plus 4th, and the 1st plus 2nd CCP, respectively (Supplementary Fig. 8). The mutants were each expressed as a Flag-tagged protein in HEK293T cells (Supplementary Figure 9). Subsequent binding analysis showed that compared to mCD46-expressing cells, mCD46ΔCCP34- and mCD46ΔCCP12-expressing cells exhibited significantly (*P* < 0.05) lower binding to C3b and CFI, respectively, while mCD46ΔCCP3- and mCD46ΔCCP34-expressing cells exhibited significantly (*P* < 0.05) lower binding to TX01 (Fig. 6c). Roughly similar binding profiles were observed with the recombinant proteins of the CCP domains of the six mCD46 variants, i.e., rmCD46ΔCCP4, rmCD46ΔCCP3, rmCD46ΔCCP2, rmCD46ΔCCP1, rmCD46ΔCCP34, and rmCD46ΔCCP12. Compared to rmCD46, rmCD46ΔCCP4, rmCD46ΔCCP3, and rmCD46ΔCCP34 exhibited significantly (*P* < 0.01) decreased binding to C3b; rmCD46ΔCCP2 and rmCD46ΔCCP12 exhibited significantly (*P* < 0.01) decreased binding to CFI; rmCD46ΔCCP3 and rmCD46ΔCCP34 exhibited significantly (*P* < 0.01) decreased binding to TX01 (Fig. 6d).

## Discussion

Previous studies have shown that bacterial outer membrane proteins can participate in serum resistance[37]. In *E. coli* K12, TraT is a cell surface-exposed protein involved in surface exclusion, serum resistance, and transporting foreign antigenic determinants to the bacterial cell surface[29,30]. In *S. typhimurium*, *traT* knockout reduced serum resistance and permeability of the outer membrane[35]. In our study, *traT*, which encodes a putative outer membrane protein, was identified as an essential gene for *E. tarda* to survive in serum. Serum incubated with the *traT* mutant, TX01ΔtraT, exhibited decreased hemolytic and bactericidal activities, implying a more consumed state of the complement system. Consistently, the amount of C5a and the chemotactic activity, caused by the cleaved products of certain complement components (such as C5a and C3a), in TX01ΔtraT-treated serum were markedly increased. These results indicate that *traT* knockout essentially disables *E. tarda* to block complement activation via the alternative pathway.

In the alternative pathway of complement activation, the C3-bound CFB is cleaved by Factor D (CFD), allowing formation of the C3 convertase complex C3Bb, which cleaves C3 to C3a and C3b[13,14]. C3b combines with C3 convertase to form the C5 convertase, which cleaves C5, resulting in the formation of the lytic MAC[13,14]. In our study, we found that TX01ΔtraT exhibited apparent binding to CFB and C5, suggesting the formation of the C3 and C5 convertases and MAC on the surface of TX01ΔtraT, which is consistent with the reduced serum survival of TX01ΔtraT. Pathogenic microorganisms can target complement regulators as a strategy to hijack host protecting functions[14,38]. In *Streptococcus* sp., *Yersinia pestis*, *Borrelia burgdorferi*, *Bacillus anthracis*, and *Candida albicans*, surface proteins that bind to CFH, CFH-related protein 1, or the complement inhibitor C4b-binding protein have been reported[28,39–47]. In our study, we found that *E. tarda* TX01 exhibited apparent binding to C3, CFH,

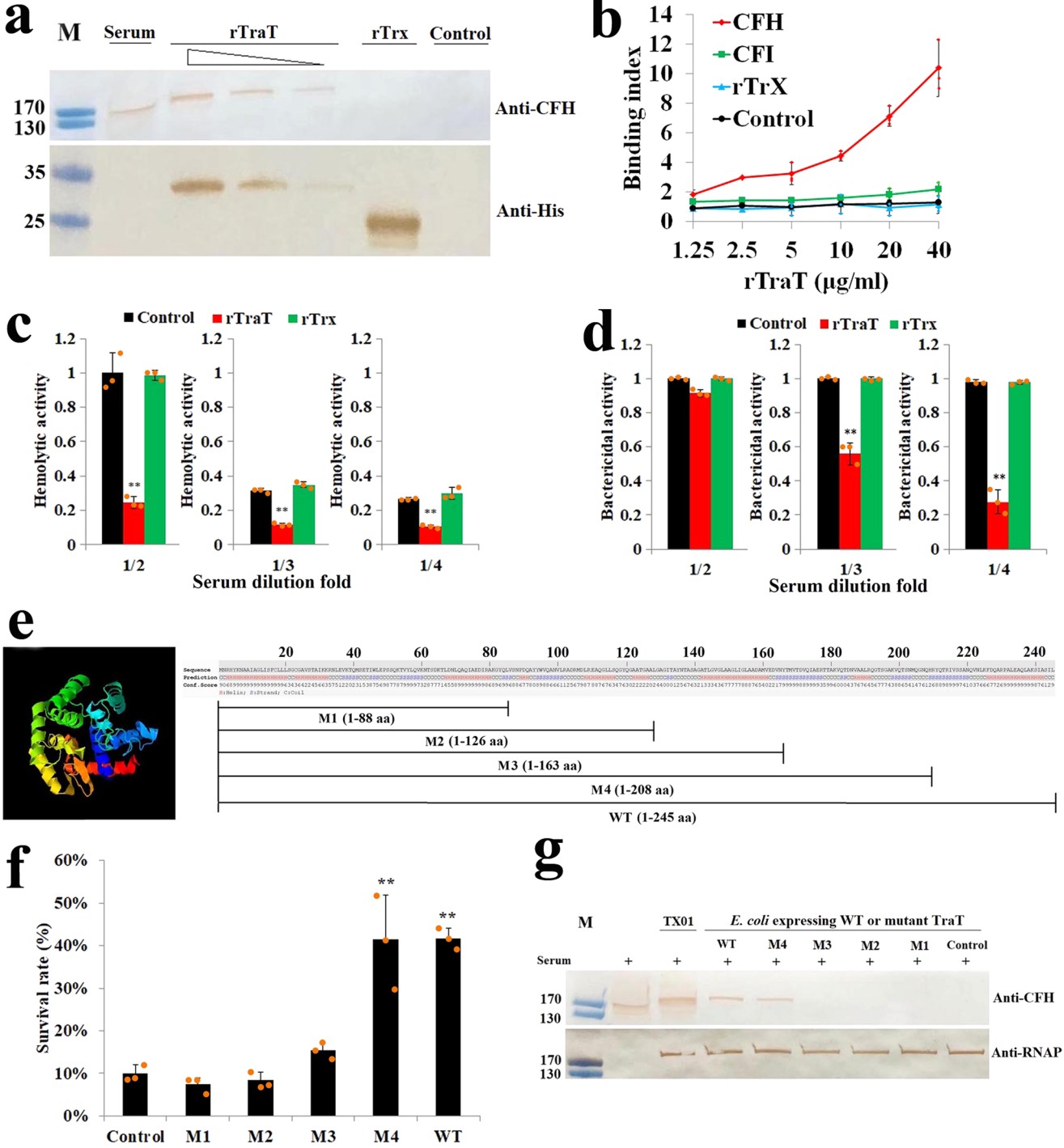

**Fig. 3 rTraT affects complement activation. a** Serum was incubated with or without (control) different concentrations of rTraT (20–80 μg/ml) or rTrx (80 μg/ml) and then subjected to IP with anti-His antibody. The immunoprecipitates were then blotted with anti-CFH and anti-His tag antibodies. M, molecular weight markers. **b** rTraT in different concentrations was incubated with or without (control) purified CFH, CFI, or rTrx (a negative protein control), and the protein bound to rTraT was determined by ELISA. **c**, **d** Serum in different dilutions was incubated with or without (control) rTraT or rTrx, and the hemolytic (**c**) and bactericidal (**d**) activities of the serum were subsequently determined. **e** The 3D structure of TraT and the schematic drawings of TraT wild type (WT) and mutants (M1 to M4). **f** *E. coli* expressing or not expressing (control) TraT wild type (WT) and mutants (M1 to M4) were treated with serum for 1 h, and bacterial survival was determined. **g** The *E. coli* variants of (**f**) as well as TX01 were treated with serum and determined for CFH binding by western blot with RNAP as a loading control. M, molecular weight markers. In panels **b**–**d**, and **f**, the data are the means of three independent experiments and presented as means ± SEM. **$**P < 0.01$.

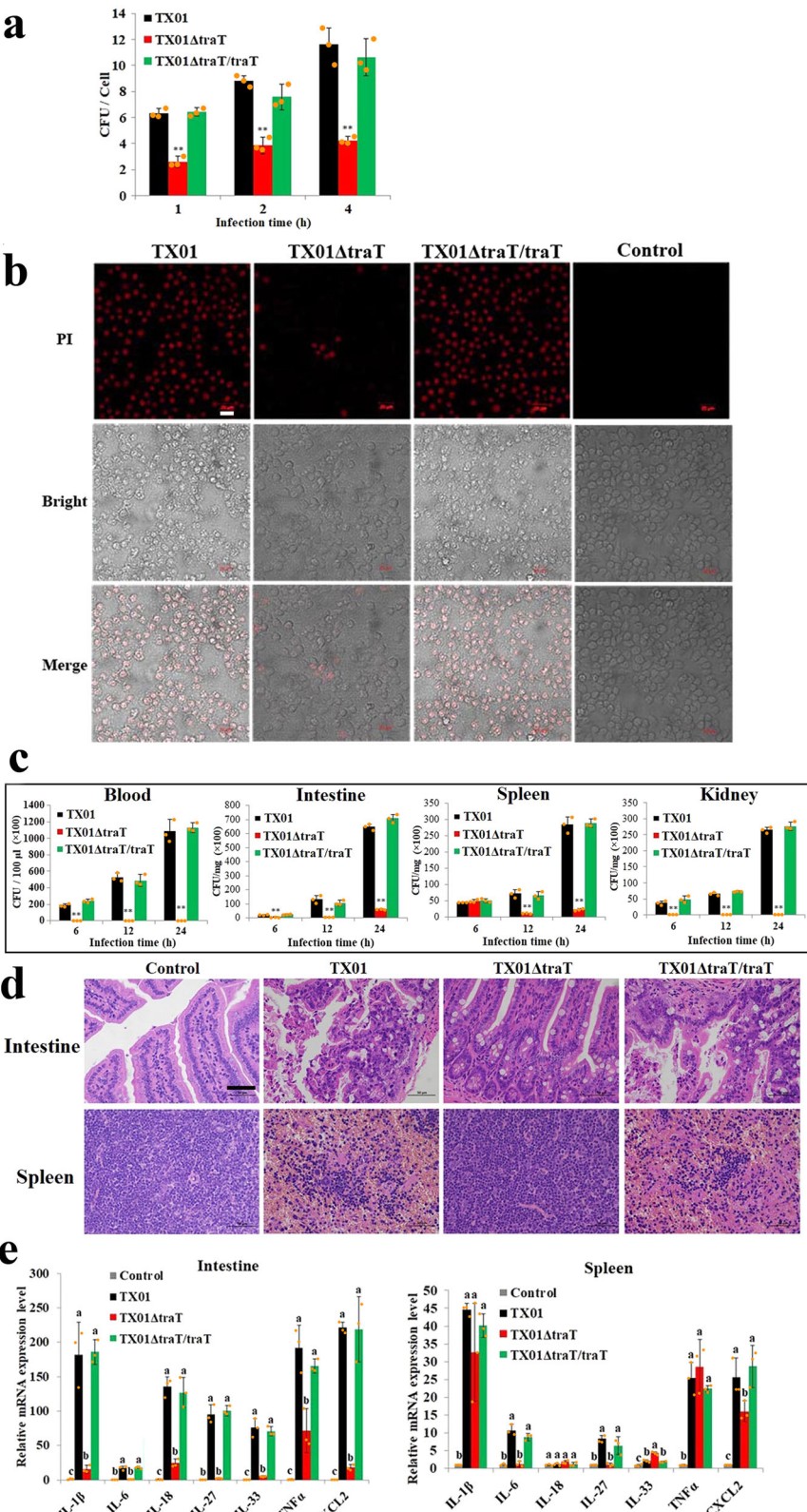

and CFI, and that depletion of CFH and CFI reduced the serum survival rate of TX01 to the level comparable to that of TX01ΔtraT. Hence, CFH and CFI are probably the key targets for TX01 to block complement activation. Compared to TX01, TX01ΔtraT displayed weakened bindings to CFH and CFI, indicating an essentialness of TraT in the recruitment of these complement factors. Consistently, rTraT bound directly to serum

CFH, as well as recombinant CFH, and inhibited complement activation. These results indicate that TraT negatively regulates complement activation by sequestering CFH from the serum and localizing it on the surface of the bacteria, which leads to subsequent recruitment of CFI and inactivation of the complement. It is interesting that in a previous report, Pramoonjago et al. showed that *E. coli* K12 TraT did not affect the activities of CFH

**Fig. 4 TraT is required for the cellular and tissue infection of *Edwardsiella tarda*. a** The ability of *E. tarda* TX01 variants to infect host cells. RAW264.7 cells were incubated with TX01, TX01ΔtraT, or TX01ΔtraT/traT for 1, 2, and 4 h. The numbers of cell-associated bacteria at each time point were determined. **b** The ability of *E. tarda* TX01 variants to induce host cell death. RAW264.7 cells were infected as above for 4 h. The control cells were uninfected. The cells were stained with PI and subjected to microscopy. Bar, 20 μm. **c** The ability of *E. tarda* TX01 variants to disseminate in host tissues. Mice were inoculated with TX01, TX01ΔtraT, or TX01ΔtraT/traT. Bacterial dissemination in tissues was determined at different hours. **d** Histopathological changes induced by *E. tarda* TX01 variants. Histopathological changes in the intestine and spleen of the mice without infection (control) and the mice infected as above for 24 h. Bar, 50 μm. **e** The effect of *E. tarda* TX01 variants on host gene expression. Mice were infected with or without (control) TX01, TX01ΔtraT, or TX01ΔtraT/traT for 12 h, and the expression of inflammatory genes in the tissues was determined by qRT-PCR. In panels a, c, data are the means of three independent assays and presented as means ± SEM. Statistical significance was determined by comparing with TX01. **$P < 0.01$. For panel e, in each gene, different letters indicate statistical significance among the samples, $P < 0.05$.

and CFI in the cleavage of C3b[34]. Compared to *E. coli* K12 TraT, *E. tarda* TraT differs by 26.5% in sequence, which may account for the distinct property of *E. tarda* TraT in complement regulation. In our study, *E. tarda* TraT expressed in *E. coli* enabled the host cells to bind CFH and survive in serum, which confirmed the CFH-dependent anti-complement activity of *E. tarda* TraT. Besides involving in complement evasion, our in vitro and in vivo infection study showed that TraT was also required for *E. tarda* to invade into and replicate in host cells, disseminate in and damage host tissues, and induce host mortality, indicating that TraT is essential not only to serum resistance, but also to the tissue dissemination and systemic infection of *E. tarda*. In addition, we also observed a participation of TraT in inflammatory response, as *traT* knockout weakened the ability of *E. tarda* to induce inflammatory gene expression in intestine and spleen, which may account for the less severe pathological conditions observed in the tissues of TX01ΔtraT-infected mice.

Outer membrane proteins are known to be involved in bacterial adhesion to host cells[48,49]. In our study, rTraT was found to bind murine macrophages by interaction with CD46. CD46 is a cofactor of CFI and facilitates CFI-mediated inactivation of the activated C3b and C4b[19,50]. CD46 is also a cell receptor for some bacterial and viral pathogens[51–53]. In our study, heterologous expression of mCD46 in human cells facilitated the cellular infection of TX01, while antibody blocking of the natural mCD46 on mouse cells impaired TX01 invasion, suggesting that the interaction between TraT and CD46 is vital to TX01 infection. In support of this, pre-treatment of mouse cells with rTraT, which precluded subsequent binding of the TraT on TX01 to cellular CD46, markedly reduced TX01 invasion into the cells. It is notable that the presence of CFH decreased the cellular attachment of TX01, suggesting that CFH and mCD46 likely bound at the same site or overlapping regions in TraT. Structurally, CD46 possesses four CCP domains[36,54,55]. Previous reports showed that CCP3 was involved in the adherence of *Neisseria gonorrhoeae* to host cells[52], and CCP2-4 interacted with C3b and C4b[56,57]. In our study, CCP2 and CCP3 single deletions decreased the binding of mCD46 to CFI and *E. tarda*, respectively, while deletion of both CCP3 and CCP4 decreased mCD46 binding to C3b. In accordance with these observations, when expressed in human cells, CCP3-4, CCP1-2, and CCP3 promoted the binding of C3b, CFI, and *E. tarda*, respectively, to the cells. These results suggest a functional specialization of the CCPs, with CCP3 and CCP4 participating primarily in the binding to CFI and TraT (*E. tarda*), respectively, while CCP3-4 participating in the binding to C3b.

In conclusion, we in this study demonstrated that *E. tarda* TraT is a key virulence factor with a dual role. As an anti-complement factor, it binds CFH in a structure-dependent manner, whereby inhibiting complement activation via the alternative pathway. As a promoter of cellular infection, it interacts with the host cell receptor CD46 in a specific CCP domain-dependent manner, whereby facilitating cellular and tissue invasion. Hence, TraT plays an important role in both

complement evasion and systematic infection of *E. tarda*. These results add insights into the mechanism of pathogenicity and immune escape of *E. tarda*.

## Methods

**Bacterial strains, plasmids, and antibodies.** Bacterial strains and plasmids used in this study are listed in Supplementary Table 1. *E. tarda* TX01, a pathogenic fish isolate, was cultured in Luria-Bertani broth (LB) at 28 °C. The *E. coli* strains were cultured in LB medium at 37 °C. Where indicated, polymyxin B, tetracycline, and chloramphenicol were supplemented at the concentrations of 50, 20, and 50 μg/ml, respectively. All antibodies were used at the concentration of 1 μg/ml.

**Animal and cell culture.** Wild-type female C57BL/6 mice (6-week-old) were purchased from Ji'nan Pengyue Laboratory Animal Breeding CO. (Jinan, China). RAW264.7 and HEK293T cell lines (Cell Resource Center, Beijing, China) were cultured in DMEM (Hyclone, South Logan, UT, USA) supplemented with fetal bovine serum (10%) (Gibco, Middleton, WI, USA), penicillin (100 U/mL) and streptomycin (100 mg/mL) at 37 °C in a 5% CO$_2$ incubator. All experiments involving live animals conducted in this study were approved by the Ethics Committee of Institute of Oceanology, Chinese Academy of Sciences (permit No. MB1911).

**Sequence analysis.** Sequence analysis was performed using the BLAST program at the National Center for Biotechnology Information (NCBI) and the Expert Protein Analysis System. Domain search was performed with the conserved domain search program of NCBI and SMART. Theoretical molecular mass and isoelectric point (pI) were predicted using Expasy compute PI/Mw tool. Multiple sequence alignment was created with DNAMAN. Subcellular localization prediction was performed with CELLO v.2.5 (http://cello.life.nctu.edu.tw/). The 3D structural figures were generated using I-TASSER of Zhang Lab (https://zhanggroup.org/I-TASSER/).

**Plasmid construction and gene mutagenesis.** To construct pETTraT, which expresses recombinant TraT (rTraT) with a His tag, the coding sequence of the extracellular region of TraT without signal peptide sequence (residues 30 to 245) was amplified by PCR with primers TraT-F/R, which were designed based on the TraT sequence (GenBank accession no. WP_015461079.1). The PCR product was ligated with the T-A cloning vector T-Simple, and the recombinant plasmid was digested with *Eco*RV to retrieve the *traT*-containing fragment, which was inserted into pET259 at the *Swa*I site, resulting in pETTraT. To construct pETmCD46, which expresses recombinant mCD46 (rmCD46) with a His tag, the coding sequence of the extracellular region of mCD46 without signal peptide (residues 43 to 328) was cloned by PCR with primers mCD46-F/R, which were designed based on the CD46 sequence (GenBank accession no. BAA31859.1). The procedures for pETmCD46 construction were as above for pETTraT construction. To construct the plasmids that express mCD46 mutants, the coding sequences of the mutants mCD46ΔCCP4, mCD46ΔCCP3, mCD46ΔCCP2, mCD46ΔCCP1, mCD46ΔCCP34, and mCD46ΔCCP12, which bear deletions of residues 239 to 294, 173 to 234, 109 to 168, 45 to 104, 173 to 294, and 45 to 168, respectively, were synthesized and inserted into pET28a by Sangon Biotech (Shanghai, China), resulting in pETmCD46 mutants. To construct pCAGGSmCD46, which expresses Flag-tagged mCD46, the coding sequence of mCD46 was amplified by PCR with primers mCD46-F1/mCD46-R1 and inserted to TA cloning vector T-Simple as above. The recombinant plasmid was digested with *Sca*I and *Nhe*I to retrieve the mCD46-coding fragment, which was inserted into plasmid pCAGGS at between the *Sca*I and *Nhe*I sites, resulting in pCAGGSmCD46. To construct the plasmids that express pCAGGSmCD46 mutants, the coding sequences of the above mCD46 mutants, i.e., mCD46ΔCCP4, mCD46ΔCCP3, mCD46ΔCCP2, mCD46ΔCCP1, mCD46ΔCCP34, and mCD46ΔCCP12, were synthesized and inserted into pCAGGS by Sangon Biotech (Shanghai, China), resulting in pCAGGSmCD46 mutants. To construct the low copy-number plasmid pJTTraT that expresses *traT*, the coding sequence of *traT* was amplified by PCR with primers TraT-F1/R1 and inserted to TA cloning vector T-Simple as above. The recombinant plasmid was digested with *Eco*RV to retrieve the *traT*-containing fragment, which was inserted

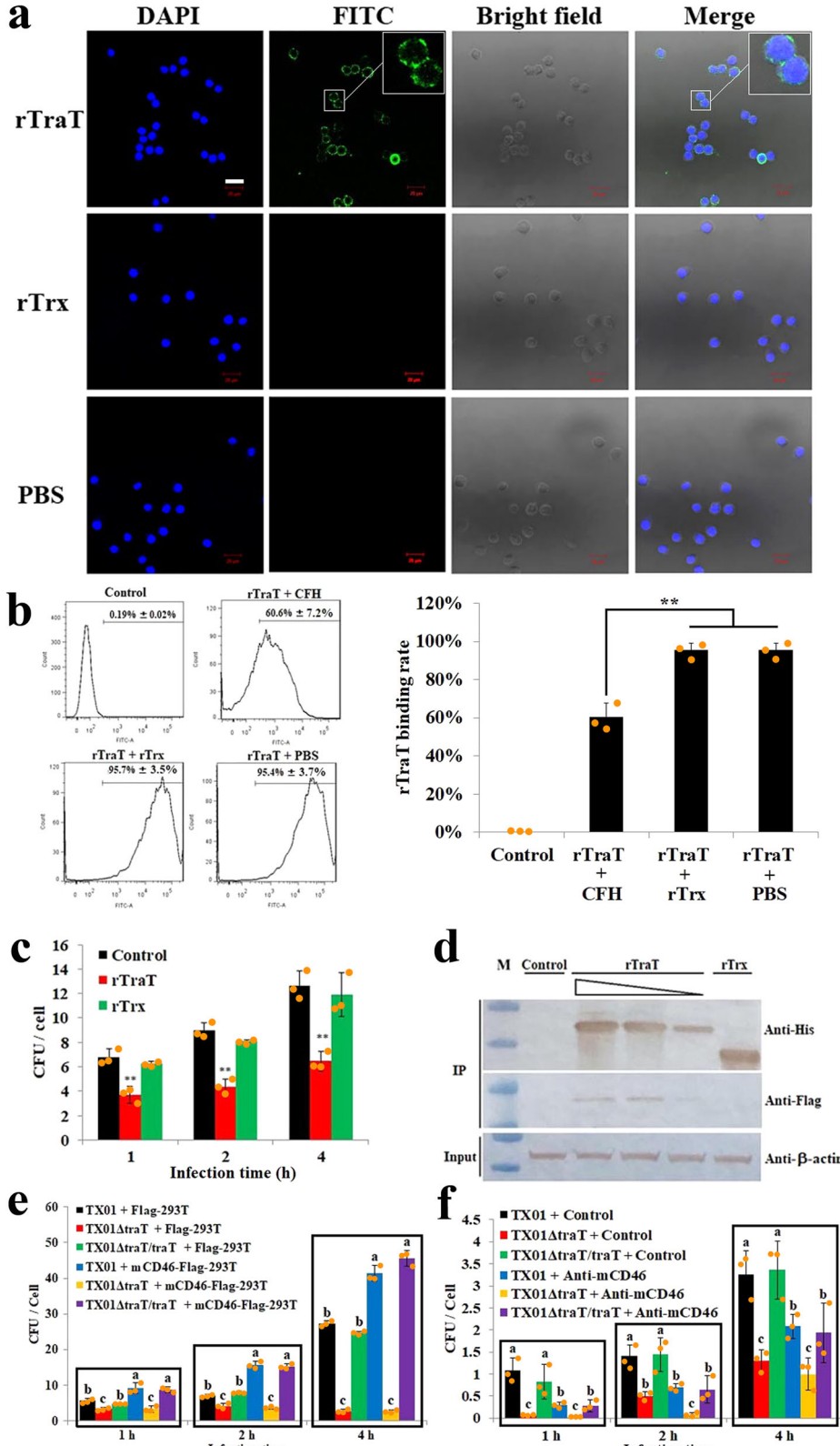

into plasmid pBT3 at the *Eco*RV site, resulting in pBT3TraT. pBT3TraT was digested with *Swa*I, and the fragment carrying *traT* was inserted into plasmid pJT at the *Swa*I site, resulting in pJTTraT. To construct the plasmids that express truncated mutants of TraT, the coding sequences of the mutant M1 (helix α1, residues 1 to 88), M2 (helix α1–α2, residues 1 to 126), M3 (helix α1–α3, residues 1 to 163), and M4 (helix α1–α4, residues 1 to 208) were generated by PCR with the primer pair TraT-F1/MR1, TraT-F1/MR2, TraT-F1/MR3, and TraT-F1/MR4, respectively. The PCR products were used to generated pJT-based plasmids

pJTTraTM1 (expressing M1), pJTTraTM2 (expressing M2), pJTTraTM3 (expressing M3), and pJTTraTM4 (expressing M4) as above. All plasmids were verified by sequence analysis. The primers for all plasmid constructions are listed in Supplementary Table 2.

**Construction of TX01ΔtraT and TX01ΔtraT/traT**. To construct *E. tarda* TX01ΔtraT, in-frame deletion of a 540 bp segment of *traT* (residues 21 to 200) was performed by overlap extension PCR as follows: the first and second overlap PCR

**Fig. 5 Binding of TraT to host cells through CD46. a** Binding of rTraT to host cells. RAW264.7 cells were incubated with rTraT, rTrx, or PBS, and cell-bound proteins were detected by fluorescence microscopy with FITC-labeled antibody. Bar, 20 μm. **b** The effect of CFH on the binding of rTraT to host cells. rTraT was pre-incubated with CFH, rTrx, or PBS, and then incubated with RAW264.7 cells. rTraT-bound cells were detected by flow cytometry and shown in graph. RAW264.7 cells alone served as a control. **c** The effect of exogenous rTraT on the infection of *Edwardsiella tarda* TX01 in host cells. RAW264.7 cells pre-treated with rTraT, rTrx, or PBS (control) were incubated with TX01 for different hours, and the number of cell-infected bacteria was determined. **d** Interaction of rTraT with mCD46. The lysate of HEK293T cells expressing Flag-tagged mCD46 was incubated with different concentrations of His-tagged rTraT (20–80 μg/ml), rTrx (80 μg/ml), or PBS (control), and then subjected to IP with anti-His antibody. The immunoprecipitate was analyzed by western blot with the antibody indicated. β-actin was used as a loading control. M, molecular weight markers. **e** The effect of mCD46 expression in human cells on the infection of TX01 variants in these cells. HEK293T cells expressing Flag tag alone (Flag-293T) or Flag-tagged mCD46 (mCD46-Flag-293T) were incubated with TX01 variants for different hours, and the number of cell-infected bacteria was determined. **f** The effect of anti-mCD46 antibody on the infection of TX01 variants in host cells. RAW264.7 cells were incubated with TX01 variants in the presence or absence (control) of anti-mCD46 antibody, and the number of cell-infected bacteria was determined at different hours. In panels b, c, e, and f, the data are the means of three independent experiments and presented as means ± SEM. **P < 0.01. For panels e, f, in each time point, different letters indicate statistical significance among the samples, P < 0.05.

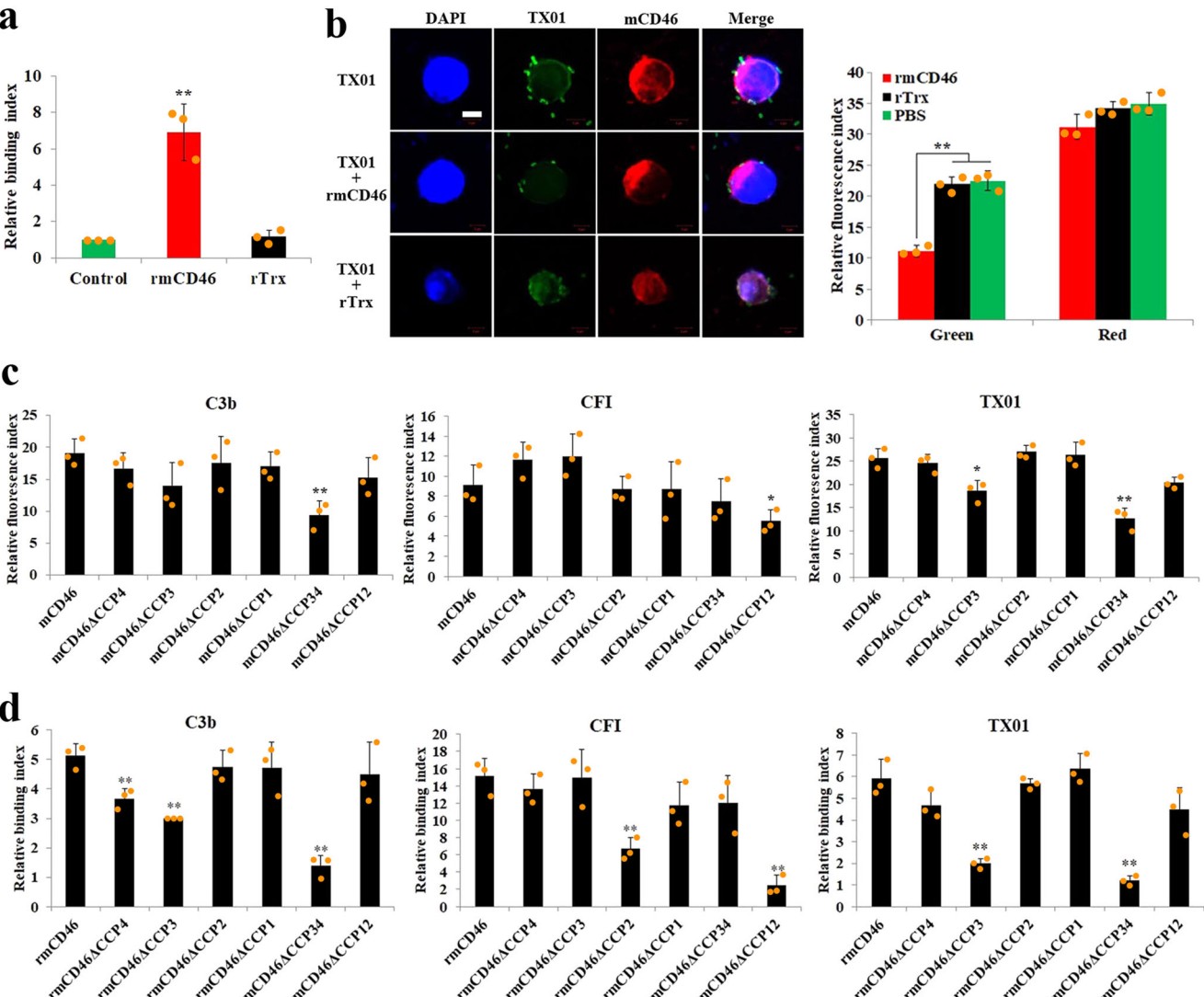

**Fig. 6 Binding of rmCD46 to *Edwardsiella tarda* and its dependence on the conserved CCP domains. a** *E. tarda* TX01 was incubated with or without (control) rmCD46 or rTrx, and the bacteria-bound protein was determined by ELISA. **b** TX01 was pre-incubated with or without rmCD46 or rTrx. The bacteria were then used to infect HEK293T cells expressing Flag-tagged mCD46. After infection, the cells were treated with Alexa 594-labeled antibody targeting Flag-tagged mCD46 and FITC-labeled antibody targeting TX01. The cells were then stained with DAPI and subjected to microscopy or measurement of green and red fluorescence. Bar, 5 μm. **c** HEK293T cells expressing Flag-tagged mCD46 or its mutants were incubated with or without C3b, CFI, or TX01 for 1 h. The cells were treated with FITC-labeled antibody targeting C3b, CFI, or TX01, and then measured for fluorescence. **d** rmCD46 and its mutants were incubated with or without C3b, CFI, or TX01, and the binding was determined by ELISA. In all panels, the data are the means of three independent experiments and presented as means ± SEM. In panels c, d, statistical significance was determined by comparing with mCD46/rmCD46. **P < 0.01; *P < 0.05.

were performed with primer pairs TraT-F2/R2 and TraT-F3/R3, respectively, and the fusion PCR was performed with the primer pair F2/R3. The final PCR product was inserted into the suicide plasmid pDM4 at the *Bgl*II site, resulting in pDMTraT. *E. coli* S17-1 λ*pir* was transformed with pDMTraT, and the transformants were conjugated with TX01 as reported previously[12]. One of the colonies resistant to sucrose and sensitive to chloramphenicol (marker of pDM4) was analyzed and confirmed to bear *traT* in-frame deletion by PCR. This strain was named TX01ΔtraT. To construct the *traT* complement strain TX01ΔtraT/traT, S17-1λpir was transformed with pJTTraT, and the transformants were conjugated with TX01ΔtraT. The transconjugants were selected on LB agar plates supplemented with tetracycline (marker of pJT) and polymyxin B (marker of TX01 and its derivatives). One of the transformants was named TX01ΔtraT/traT.

**Purification of recombinant proteins and preparation of antibody.** *E. coli* BL21 (DE3) was transformed separately with pETTraT, pETmCD46, pETmCD46 mutants, and pET32a (which expresses the Trx tag). The transformants were cultured in LB medium at 37 °C to mid-log phase, and the expression of the recombinant proteins (rTraT, rmCD46, rmCD46 mutants, and rTrx) was induced by adding isopropyl-β-D-thiogalactopyranoside to a final concentration of 1 mM. After growth at 16 °C for an additional 16 h, the cells were harvested by centrifugation, and recombinant proteins were purified using nickel-nitrilotriacetic acid columns (GE Healthcare, Piscataway, NJ, USA) as recommended by the manufacturer. The purified proteins were treated with Triton X-114 to remove endotoxin as reported previously[58]. The proteins were dialyzed for 24 h against phosphate-buffered saline (PBS) and concentrated using PEG 20000. The concentrations of the purified proteins were determined using NanoPhotometer (Implen GmbH, Munich, Germany). Mouse antibodies against rTraT and rTrx were prepared as reported previously[59] and purified using rProtein G Beads (Solarbio, Beijing, China).

**Analysis of *E. tarda* serum resistance.** Serum survival assay was performed as reported previously[5]. Briefly, *E. tarda* strains were cultured in LB medium to an OD$_{600}$ of 0.8. The cells were washed with PBS and resuspended in PBS. Approximately $10^5$ bacterial cells were mixed with 50 μl untreated or heat-inactivated (control) mouse serum. After incubation with mild agitation at 37 °C for 1 h, the mixture was serially diluted and plated in triplicate on LB agar plates. The plates were incubated at 28 °C for 48 h, and the colonies were enumerated. The genetic identity of the colonies was verified as reported previously[5]. The survival rate was calculated as follows: (number of bacteria surviving serum treatment/ number of bacteria surviving heat-inactivated serum treatment) × 100%. To examine the serum survival of *E. coli* expressing TraT variants, *E. coli* BL21 was transformed with pJTTraT, pJTTraTM1, pJTTraTM2, pJTTraTM3, pJTTraTM4, and pJT. The transformants were cultured in LB medium to an OD$_{600}$ of 0.8 at 37 °C and resuspended to $2 \times 10^6$ CFU/ml in PBS. The bacterial suspension was mixed with heated or unheated mouse serum (1/8 dilution), followed by incubation at 37 °C for 1 h. Serum survival rate was determined as above. To examine the effect of CFH or CFI depletion on TraT-depended inhibition of complement activation, mouse serum was treated with anti-CFH antibody or anti-CFI antibody (Abcam, Cambridge, MA, USA) at 4 °C for 4 h. After incubation, the mixture was transferred to the spin column bound with protein-A/G and incubated at 4 °C for 2 h. The column was then centrifuged, and the serum was collected. Serum survival of the bacteria was determined as above. To examine the effect of serum on the cellular morphology of *E. tarda*, *E. tarda* was cultured as above and resuspended in PBS to $10^8$ CFU/ml. The cells were incubated with or without (control) mouse serum at 37 °C for 1 h and then observed with a transmission electron microscope (HT7700, Hitachi, Japan).

**Hemolytic, bactericidal, and chemotactic activities of bacteria-treated serum.** To prepare bacteria-treated serum, *E. tarda* variants and *E. coli* DH5a were cultured and resuspended to $10^9$ CFU/ml in HBSS (Solarbio, Beijing, China) as above. Mouse serum was diluted two times in HBSS and mixed with an equal volume of bacterial suspension or HBSS. The mixture was incubated at 37 °C for 1 h. The serum was then diluted and passed through a 0.22 mM filter to remove any bacterial cells. The serum was used for the assays of hemolysis, bactericidal activity, and chemotaxis as reported previously[5,60]. C5a concentration in the serum was determined with mouse C5a ELISA kit (ELISAGenis, Oxfordshire, UK) as recommended by the manufacturer.

**Immunoblot assay.** *E. tarda* TX01, TX01ΔtraT, TX01ΔtraT/traT, and *E. coli* DH5a were cultured and resuspended to $2 \times 10^9$ CFU/ml in PBS as above. Bacteria were incubated with or without mouse serum at 37 °C for 1 h. The bacteria were washed with PBS three times, and whole-cell proteins were prepared and subjected to immunoblotting as reported previously[61] with antibodies against C3, C5, CFB, CFH, and CFI (Abcam, Cambridge, MA, USA). RNA polymerase beta was used as a loading control[62] and detected with anti-RNA polymerase beta antibody (Abcam, Cambridge, MA, USA).

**Immunoprecipitation.** To examine the interaction between rTraT and serum CFH, mouse serum-containing rTraT (20, 40, or 80 μg/ml), rTrx (80 μg/ml), or PBS

(control) was incubated at 28 °C for 1 h and then treated with anti-His antibody at 4 °C for overnight. The mixture was subjected to immunoprecipitation using an Immunoprecipitation Protein A/G Plus Agarose Kit (Sangon Biotech, Shanghai, China) according to the manufacturer's instructions. The immunoprecipitated complex and serum (positive control) were resolved by SDS-PAGE. The proteins were transferred to a nitrocellulose membrane and immunoblotted as above with anti-CFH antibody or anti-His antibody.

**SPR assay.** SPR assay was performed according to the method of a previous report[63]. CFH protein was immobilized using amine-coupling chemistry on the surface of a sensor chip with a modified alginate-based polymer matrix bound to a gold layer. The binding of CFH to rTraT (80, 40, 20, 10, 5, or 2.5 μg/ml), rTrx (80 μg/ml), or PBS (control) was then detected as reported previously[63].

**In vivo infection.** In vivo infection was performed as reported previously[12]. Briefly, *E. tarda* TX01, TX01ΔtraT, and TX01ΔtraT/traT were cultured as above. The cells were washed with PBS and resuspended in PBS to $2.5 \times 10^8$ CFU/ml. Mice (average 15.7 g) were randomly divided into three groups (9 mice/group) and infected via tail vein injection with 200 μl TX01, TX01ΔtraT, or TX01ΔtraT/traT. At 12, 24, and 48 hpi, blood, intestine, spleen, and kidney were collected from the mice (three at each time point). The tissues were homogenized in PBS. The homogenates was serially diluted and plated in triplicate on LB agar plates. The plates were incubated at 28 °C for 48 h, and the appeared colonies were enumerated. The genetic identities of the colonies were verified by PCR and sequencing. For mortality analysis, three groups of mice (10 animals/group) were infected as above and they monitored for mortality for 48 h. For histopathological analysis, tissues were prepared from the mice at 24 hpi and immersed in 4% paraformaldehyde. Tissue sectioning and HE staining were performed by Wuhan Servicebio Technology Co., Ltd. (Wuhan, China). After dewaxing, hematoxylin staining, eosin staining, and dehydration, the tissue sections were observed with an optical microscope (Eclipse E100, Nikon, Japan). For inflammatory gene expression, total RNA was prepared from the tissues of the mice at 12 hpi with RNA-easy isolation reagent and used for quantitative real-time PCR (qRT-PCR). The PCR primers are listed in Supplementary Table 2.

**In vitro infection.** For bacterial infection of RAW264.7 cells, the cells cultured in 96-well plates ($10^5$ CFU/well) were infected with *E. tarda* TX01, TX01ΔtraT, or TX01ΔtraT/traT at a multiplicity of infection (MOI) of 5:1 as described previously[64]. The cells were incubated at 28 °C for 1, 2, or 4 h. After incubation, the plates were washed with PBS, and the cells were lysed with 100 μl PBS containing 1% Triton X-100. The cell lysate was diluted and plated in triplicate on LB agar plates. The plates were incubated at 28 °C for 48 h, and the merged colonies were counted. The identity of the colonies was verified as described above. To examine the effect of anti-mCD46 antibody on bacterial infection, the cells were pre-incubated with or without anti-mCD46 antibody (Abcam, Cambridge, MA, USA) at 37 °C for 1 h. The cells were then treated with *E. tarda* and examined for cellular infection as above. To examine cell death, RAW264.7 cells were infected with *E. tarda* at MOI of 5:1 as above. At 4 hpi, the cells were immobilized with 4% paraformaldehyde at room temperature for 0.5 h and stained with PI (20 μg/ml) for 20 min. The cells were visualized with a confocal microscope (Carl Zeiss, Oberkochen, Germany). For bacterial infection of HEK293T cells expressing mCD46, HEK293T cells were seeded in 96-well plates ($1 \times 10^4$ cells per well) overnight. The cells were transfected with or without the plasmid pCAGGSmCD46 or pCAGGS (0.1 μg per well) expressing Flag-tagged mCD46 or Flag tag, followed by incubation in fresh Opti-MEM medium for 24 h. The cells were then treated with *E. tarda* TX01, TX01ΔtraT, or TX01ΔtraT/traT for 1, 2, and 4 hpi as above, and the number of cell-infected bacteria was determined as above. To examine the effect of rmCD46 on *E. tarda* infection of mCD46-expressing HEK293T cells, *E. tarda* TX01 was pre-incubated with or without 40 μg/ml rmCD46 or rTrx for 2 h at 28 °C. HEK293T cells were transfected with pCAGGSmCD46 or pCAGGS as above and then infected with the pre-incubated TX01 at MOI of 1:10 for 2 h at 28 °C. The cells were immobilized with 4% paraformaldehyde at room temperature for 0.5 h, and anti-TX01 antibody and anti-Flag antibody (Abconal, Wuhan, China) were added into cells. The cells were incubated for 1 h at 37 °C. FITC-labeled or Alexa 594-labeled second antibodies (Abcam, Cambridge, MA, USA) were added into cells and incubated for 1 h at 37 °C. The cells were stained with DAPI for 10 min and washed with PBS for three times. The cells were observed with a confocal microscope (Carl Zeiss, Oberkochen, Germany).

**Binding of rTraT to RAW264.7 cells.** RAW264.7 cells were incubated with rTraT (40 μg/ml), rTrx (40 μg/ml), or PBS for 2 h at 28 °C. After washing with PBS, the cells were immobilized with 4% paraformaldehyde at room temperature for 0.5 h. Then, the cells were blocked with 6% skim milk for 2 h at 37 °C. Mouse anti-His antibody and FITC-labeled goat anti-mouse IgG (Abcam, Cambridge, MA, USA) were added into cells. The cells were incubated for 1 h at 37 °C and stained with DAPI for 10 min. The cells were washed with PBS for three times and observed with a confocal microscope. To determine the effect of CFH on the cellular binding of rTraT, rTraT (40 μg/ml) was incubated with or without CFH (20 μg/ml), rTrx (20 μg/ml), or PBS for 2 h at 28 °C. The mixture was then incubated with

RAW264.7 cells for 2 h at 28 °C and then washed three times with PBS. Mouse anti-His antibody and FITC-labeled goat anti-mouse IgG were added into the cells. The cells were washed with PBS and analyzed by flow cytometry with a FACSAria II flow cytometer (Becton Dickinson Biosciences, NJ, USA).

**Binding of mCD46 variants to C3b, CFI, and *E. tarda*.** To examine the binding of mCD46 and its mutants to HEK293T cells, the cells cultured in 96-well plates ($1 \times 10^4$ cells per well) were transfected as above with pCAGGSmCD46 or pCAGGSmCD46 mutants that expresses Flag-tagged mCD46 or mCD46 mutants. The cells were then incubated with *E. tarda* ($10^5$ CFU/well), CFI (20 μg/ml), or C3b (20 μg/ml) at 37 °C for 1 h and immobilized with 4% paraformaldehyde at room temperature for 0.5 h. Then, the cells were blocked with 6% skim milk for 2 h at 37 °C. Antibodies against C3b, CFI, or *E. tarda* TX01 was added to cells. The cells were incubated at 37 °C for 1 h. FITC-labeled second antibodies were added into cells. After incubation at 37 °C for 1 h, cell-bound protein/bacteria was determined by measuring the fluorescence using Multi-Mode Microplate Reader (Tecan, Mannedorf, Switzerland). To examine the interaction between rmCD46 variants and *E. tarda*/CFI/C3b, *E. tarda* suspensions ($10^8$ CFU/well), CFI (20 μg/ml), or C3b (20 μg/ml) was placed into ELISA plates and incubated at 4 °C for overnight. Skim milk powder (6%) in PBS was added to the plates, and the plates were incubated at 37 °C for 1 h. The plates were washed three times with PBST (PBS containing 0.05% Tween20), and 100 μl of rmCD46 (40 μg/ml), each of the rmCD46 mutants (40 μg/ml), or PBS was added to the plates. The plates were incubated at 37 °C for 2 h and washed as above. Mouse anti-His antibody was added to the plate. After incubation at 37 °C for 1 h, ELISA was performed as reported previously[65].

**Statistics and reproducibility.** All experiments were performed three times. Statistical analyses were carried out with SPSS 17.0 software (SPSS Inc., Chicago, IL, USA). Data were analyzed with analysis of variance (ANOVA), and statistical significance was defined as $P < 0.05$.

**Reporting summary.** Further information on research design is available in the Nature Research Reporting Summary linked to this article.

## Data availability
All relevant data are included in the article or its supplementary information file. The uncropped images of the western blots are provided in Supplementary Fig. 10. The Addgene ID numbers of newly generated plasmids are provided in Supplementary Table 1. The source data are provided in Supplementary Data 1.

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

## Acknowledgements

This work was supported by the grants from the National Key Research and Development Program of China (2018YFD0900500), the National Natural Science Foundation of China (31972831 and 41906108), the Young Scholar Program of Tianjin Distinguished Professor, and the Taishan Scholar Program of Shandong Province.

## Author contributions

M.F.L. and L.S. conceived and designed the experiments, M.F.L., M.W., and Y.Y.S. performed the experiments and analyzed the data, M.F.L. wrote the manuscript, L.S. edited the manuscript.

## Competing interests

The authors declare no competing interests.
