## [Peer Review File · Communications Biology]

Reviewers' comments:

Reviewer #1 (Remarks to the Author):

The authors' group has earlier found that *E. tarda* can significantly resist serum killing, and found that a variety of *E. tarda* virulence factors are involved in this process. In this study, the authors found that *E. tarda* TraT plays a key role in resistance to serum killing and elucidated the mechanism of action. In addition, the authors further revealed that *E. Tarda* TraT also plays an important role in promoting cell infection and tissue diffusion. This work provides important support for revealing the immune escape mechanism of *E. tarda*, and also provides important clues for understanding the mechanism of Anti-complement killing mediated by TraT.

At the same time, there are some major shortcomings in the research work, which need to be improved or revised.

1. In Fig1B, I suggested that the electron microscope data of *E. coli* with and without serum should be given as reference. Besides, are the *E. coli* data not included in Fig1E and F?
2. In Fig2A and B, there are obvious differences in C5 and CFB, why are they two not analyzed or further discussed?
3. In Fig3C and D, after the co-incubation of rTraT with serum, CFH in serum should be exhausted. After the depletion of CFH in serum, the negative regulation of complement killing should be blocked, and thus its complement killing activity should be increased. But, the Fig3C and D showed that hemolytic activity and bactericidal activity were decreased, why? please give reasonable explanations.
4. TraT is also present in *E. coli*, which may not affect the regulation of CFH and CFI activities as discussed in line 221-223. In order to strengthen the universality of TraT mechanism revealed in this study, I suggested to supplement *E. coli* traT deletion mutant as controls in some important experiments.
5. I suggested to add enlarged vision picture with more details of the positive field of in Fig5A.

Reviewer #2 (Remarks to the Author):

Li and colleagues have identified the *Edwardsiella tarda* outer membrane protein TraT as a recruiter of Factor H and promoter of CD46-mediated cellular invasion. This dual role of complement inhibition and cellular infection is believed to significantly contribute to systemic infection as deletion of traT results in less mortality, less bacterial dissemination in tissues in a mouse model of infection and corresponding decrease in expression of important inflammatory cytokines.

The authors have shown that TraT is an important gene in resisting complement killing by recruiting the negative complement regulator Factor H. Interaction between TraT and FH has been shown via western blotting and ELISA. FH depleted serum and Factor I depleted serum resulted in no significant difference in survival between WT and traT deleted strains whereas significant survival was observed in normal human serum when comparing WT and traT mutant. In addition, using recombinantly produced traT mutants, the first four helices of TraT were found to be important for FH interaction. Deletion of TraT resulted in less severe infection using a murine model. Moreover, a second function of TraT was analysed whereby TraT was shown to interact with CD46 promoting cellular invasion. The CCP3-4 of CD46 have been shown to be important for TraT interaction.

The manuscript is well written, including experimental details and conclusions drawn are logical and will be of interest to the field.

I offer some comments and suggestions on the manuscript:

Discussion points:

- When rTraT is incubated in serum, we see a reduction in haemolytic and bactericidal activity (Fig 3C-D). As rTraT binds FH, it is unclear why this would lead to less activity in these assays unless TraT recruits FH to the RBC and bacteria surface and limits complement activation in this fashion.

Is this the case or does TraT have another function to inactivate complement?

- What is the affinity between TraT and FH – kD values derived using solid-phase ELISA or more sensitive methods such as surface plasmon resonance would indicate the relative strength of this interaction.
- What CCP domains of FH are recruited – many studies have been able to determine the exact CCP domains of FH that microbes bind to recruit to their surface for complement evasion. Using mutants or anti-Factor H monoclonal antibodies that target specific CCP domains could be used to assess what domains of FH are bound by TraT.

Minor:

- 1) The numbering representing statistical significance is not clear – Apologies if I have missed where this is explained – Perhaps this can be included in the figure legends
- 2) In Figure 1, there is not E coli data shown for panel E and F however the panel indicates the existence of an E coli control.

Reviewer #3 (Remarks to the Author):

The manuscript by Li. et. al investigates the role of one gene, TraT in the pathogenesis of infection by *Edwardsella tarda*. The major findings of this paper focus on the role of TraT in inhibition of the complement cascade, as well as specific binding to host cell receptors during infection.

The overall impression from reading this paper is that of a thorough and detailed study that has investigated the role of TraT as a virulence factor from multiple angles. My main criticisms of the paper are in clarity of writing, as I find the data presented to be convincing with appropriate controls, once I teased out the design of each experiment.

My suggestions to improve this paper are as follows:

- 1) More information as to the experimental design for the figures needs to be presented in the results section and/or the figure legends for the entire paper. Examples of places where more detail is needed include:
 - a. Figure 2 in particular, it is very difficult to tease apart what samples are being tested (bacteria or bacterial treated sera, etc.) and what is being probed.
 - b. More explicit labels of figures would help in this regard. Figure 2A would have much enhanced clarity if it labelled the specific antigen/antibody combo being probed rather than just 2Aa, 2Ab etc.
 - c. A number of the figures include letters to designate statistical significance. However, the figure legends need to explicitly explain the comparison being conducted (WT vs KO etc.) for each letter, and the p value.
 - d. The results text needs to include the mode of infection for in vivo experiments
 - e. Figure 4B and 5B it is unclear what is being tested.
- 2) More information is needed about the controls for experiments where an arbitrary binding index was used as the output. Please explain positive and negative controls, normalization etc.
- 3) Headers of section should spell out all components, rather than using acronyms such as CFH.
- 4) The discussion should include more about what is known about the function of TraT in *E. coli* and other bacteria- does it have a function other than virulence factor?

Reviewer #1 (Remarks to the Author):

The authors' group has earlier found that *E. tarda* can significantly resist serum killing, and found that a variety of *E. tarda* virulence factors are involved in this process. In this study, the authors found that *E. tarda* TraT plays a key role in resistance to serum killing and elucidated the mechanism of action. In addition, the authors further revealed that *E. Tarda* TraT also plays an important role in promoting cell infection and tissue diffusion. This work provides important support for revealing the immune escape mechanism of *E. tarda*, and also provides important clues for understanding the mechanism of Anti-complement killing mediated by TraT.

At the same time, there are some major shortcomings in the research work, which need to be improved or revised.

1. In Fig1B, I suggested that the electron microscope data of *E. coli* with and without serum should be given as reference. Besides, are the *E. coli* data not included in Fig1E and F?

Reply:

- (1) According to reviewer's suggestion, *E. coli* was included in Figure 1 B as a reference.
- (2) We are very thankful to the reviewer for indicating the omission of the *E. coli* data in Figure 1E and F. The *E. coli* data were added to the revised Figure 1.

2. In Fig2A and B, there are obvious difference in C5 and CFB, why are they two not analyzed or further discussed ?

Reply: The difference in C5 and CFB was indicated in the Results (lines 106-108). According to reviewer's suggestion, the relevant discussion was added to the revised manuscript. Lines 213-218.

3. In Fig3C and D, after the co-incubation of rTraT with serum, CFH in serum should be exhausted. After the depletion of CFH in serum, the negative regulation of complement killing should be blocked, and thus its complement killing activity should be increased. But, the Fig3C and D showed that hemolytic activity and bactericidal activity were decreased, why? please give reasonable explanations.

Reply: As shown in Figure 2F, the wild type TX01 binds both CFH and CFI, and the binding to CFI occurs only after CFH binding, indicating that it is the TraT-CFH complex that recruits CFI. In addition, TraT may also interact with C3, since, as shown in Figure 2A and B, TX01 binds significantly more C3 than TX01ΔtraT. In Figure 3C and D, the presence of rTraT allows the association of CFH with CFI, thus activating CFI; rTraT may, through its interaction with C3, also promote the binding of the CFH-CFI complex to rabbit red blood cells (RBCs) (in hemolytic assay) and *E. coli* (in bactericidal assay), thus facilitating cleavage of the C3b on RBCs/*E. coli* by the activated CFI, resulting in inhibition of complement activation.

4. TraT is also present in *E. coli*, which may not affect the regulation of CFH and CFI activities as discussed in line 221-223. In order to strengthen the universality of TraT mechanism revealed in this study, I suggested to supplement *E. coli* traT deletion mutant as controls in some important experiments.

Reply: The *E. coli* strain used in the TraT studies is *E. coli* K12 (Pramongjago et al., 1992; Binns et al., 1982). In our study, the *E. coli* strain used is *E. coli* DH5 α . The genome of DH5 α does not have the *traT* gene. The lack of *traT* in DH5 α was confirmed by PCR analysis, which did not detect the presence of the *traT* gene. In the revised manuscript, “K12” was added to the appropriate places. Lines 72 230, and 231.

5. I suggested to add enlarged vision picture with more details. The positive field of in Fig5A.

Reply: According to reviewer’s suggestion, enlarged vision pictures of the positive field were added to Figure 5A.

Reviewer #2 (Remarks to the Author):

Li and colleagues have identified the *Edwardsiella tarda* outer membrane protein TraT as a recruiter of Factor H and promoter of CD46-mediated cellular invasion. This dual role of complement inhibition and cellular infection is believed to significantly contribute to systemic infection as deletion of *traT* results in less mortality, less bacterial dissemination in tissues in a mouse model of infection and corresponding decrease in expression of important inflammatory cytokines.

The authors have shown that TraT is an important gene in resisting complement killing by recruiting the negative complement regulator Factor H. Interaction between TraT and FH has been shown via western blotting and ELISA. FH depleted serum and Factor I depleted serum resulted in no significant difference in survival between WT and *traT* deleted strains whereas significant survival was observed in normal human serum when comparing WT and *traT* mutant. In addition, using recombinantly produced *traT* mutants, the first four helices of TraT were found to be important for FH interaction. Deletion of TraT resulted in less severe infection using a murine model. Moreover, a second function of TraT was analysed whereby TraT was shown to interact with CD46 promoting cellular invasion. The CCP3-4 of CD46 have been shown to be important for TraT interaction.

The manuscript is well written, including experimental details and conclusions drawn are logical and will be of interest to the field.

I offer some comments and suggestions on the manuscript:

Discussion points:

- When rTraT is incubated in serum, we see a reduction in haemolytic and bactericidal activity (Fig 3C-D). As rTraT binds FH, it is unclear why this would lead to less activity in these assays unless TraT recruits FH to the RBC and bacteria surface and limits complement activation in this fashion. Is this the case or does TraT have another function to inactivate complement?

Reply: As shown in Figure 2F, the wild type TX01 binds both CFH and CFI, and the binding to CFI occurs only after CFH binding, indicating that it is the TraT-CFH complex that recruits CFI. In addition, TraT may also interact with C3, since, as shown in Figure 2A and B, TX01 binds significantly more C3 than TX01 Δ traT. In Figure 3C and D, the presence of rTraT allows the association of CFH with CFI, thus activating CFI; rTraT may, through its interaction with C3, also promote the binding of the CFH-CFI complex to rabbit red blood cells (RBCs) (in hemolytic assay) and *E. coli* (in bactericidal assay), thus facilitating cleavage of the C3b on RBCs/*E. coli* by the activated CFI, resulting in inhibition of complement activation.

- What is the affinity between TraT and FH – kD values derived using solid-phase ELISA or more sensitive methods such as surface plasmon resonance would indicate the relative strength of this interaction.

Reply: According to reviewer's suggestion, we examined the binding affinity between rTraT and CFH by surface plasmon resonance. The relevant result and method were added to the revised manuscript. Lines 126-128 and 396-399, Supplementary Fig. 2.

- What CCP domains of FH are recruited – many studies have been able to determine the exact CCP domains of FH that microbes bind to recruit to their surface for complement evasion. Using mutants or anti-Factor H monoclonal antibodies that target specific CCP domains could be used to assess what domains of FH are bound by TraT.

Reply: We appreciate the reviewer's suggestion. However, since CFH is a large protein (~139 kDa), it is difficult to obtain the recombinant protein of CFH, either mutant or wildtype. For antibodies, during this study, we purchased a number of commercial CFH antibodies, including ab8842, ab180545, ab133536, and ab118820, however, only ab8842 proved to be effective in detecting the CFH in mouse serum by Western blot. For this reason, ab8842 was the antibody used in our study.

Minor:

- 1) The numbering representing statistical significance is not clear – Apologies if I have missed where this is explained – Perhaps this can be included in the figure legends

Reply: Statistical significance was defined as $P < 0.05$. The relevant description was added to the legends of Figures 1, 2, 4 and 5 in the revised manuscript.

- 2) In Figure 1, there is not *E. coli* data shown for panel E and F however the panel indicates the existence of an *E. coli* control.

Reply: We are very thankful to the reviewer for indicating the omission of the *E. coli* data in Figure 1E and F. The *E. coli* data were added to the revised Figure 1.

Reviewer #3 (Remarks to the Author):

The manuscript by Li. et. al investigates the role of one gene, TraT in the pathogenesis of infection by *Edwardsella tarda*. The major findings of this paper focus on the role of TraT in inhibition of the complement cascade, as well as specific binding to host cell receptors during infection.

The overall impression from reading this paper is that of a thorough and detailed study that has investigated the role of TraT as a virulence factor from multiple angles. My main criticisms of the paper are in clarity of writing, as I find the data presented to be convincing with appropriate controls, once I teased out the design of each experiment.

My suggestions to improve this paper are as follows:

1) More information as to the experimental design for the figures needs to be presented in the results section and/or the figure legends for the entire paper. Examples of places where more detail is needed include:

a. Figure 2 in particular, it is very difficult to tease apart what samples are being tested (bacteria or bacterial treated sera, etc.) and what is being probed.

Reply: According to reviewer's suggestion, more information was added to Figure 2 and the legends of Figures 2, 4, and 5.

b. More explicit labels of figures would help in this regard. Figure 2A would have much enhanced clarity of it labelled the specific antigen/antibody combo being probed rather than just 2Aa, 2Ab etc.

Reply: Figure 2A was revised according to the reviewer's suggestion.

c. A number of the figures include letters to designate statistical significance. However, the figure legends need to explicitly explain the comparison being conducted (WT vs KO etc.) for each letter, and the p value.

Reply: Different letters indicate significant differences among the samples ($P < 0.05$). The relevant description was added to the legends of Figures 1, 2, 4 and 5 in the revised manuscript.

d. The results text needs to include the mode of infection for in vivo experiments

Reply: The mode of infection was added to the revised manuscript. Line 146.

e. Figure 4B and 5B it is unclear what is being tested.

Reply: Relevant descriptions were added to the figure legends of Figure 4B and 5B.

2) More information is needed about the controls for experiments where an arbitrary binding index was used as the output. Please explain positive and negative controls, normalization etc.

Reply: In Figure 3B, PBS was the negative control, and rTrx, which was prepared under the same condition as rTraT, was used a negative protein control for rTraT. Figure 3B legend was modified.

3) Headers of section should spell out all components, rather than using acronyms such as CFH.

Reply: The reviewer's suggestion was taken. Lines 103, 119, and 176.

4) The discussion should include more about what is known about the function of TraT in *E. coli* and other bacteria- does it have a function other than virulence factor?

Reply: More discussion on the function of TraT in *E. coli* and other bacteria was added according to the reviewer's suggestion. Lines 203-205.

REVIEWERS' COMMENTS:

Reviewer #1 (Remarks to the Author):

This revised manuscript has basically addressed most of my concerns, i don't have further comments.

Reviewer #2 (Remarks to the Author):

The authors state that they have performed SPR, examining the binding affinity between rTraT and CFH. While I can see the data in the Suppl Figure 2, the authors appear to have only used one concentration of rTraT and therefore have not calculated any K_D value or commented on the relative binding between the two proteins.

Apart from the above, authors have addressed all of my concerns.

Reviewer #3 (Remarks to the Author):

The authors have addressed all concerns presented in the first set of reviews.

Reviewer #2 (Remarks to the Author):

The authors state that they have performed SPR, examining the binding affinity between rTraT and CFH. While I can see the data in the Suppl Figure 2, the authors appear to have only used one concentration of rTraT and therefore have not calculated any K_D value or commented on the relative binding between the two proteins.

Reply: According to reviewer's suggestion, we examined the binding affinity between rTraT and CFH. The relevant result was added to the revised manuscript. Lines 126-127 and 398-399, Supplementary Figure 2.